# Omnigrasp: Grasping Diverse Objects with Simulated Humanoids

Zhengyi Luo[1,2] *    Jinkun Cao[1] *    Sammy Christen[2,3]    Alexander Winkler[2]
Kris Kitani[1,2] †    Weipeng Xu[2] †

[1]Carnegie Mellon University; [2]Reality Labs Research, Meta; [3]ETH Zurich
https://zhengyiluo.github.io/Omnigrasp

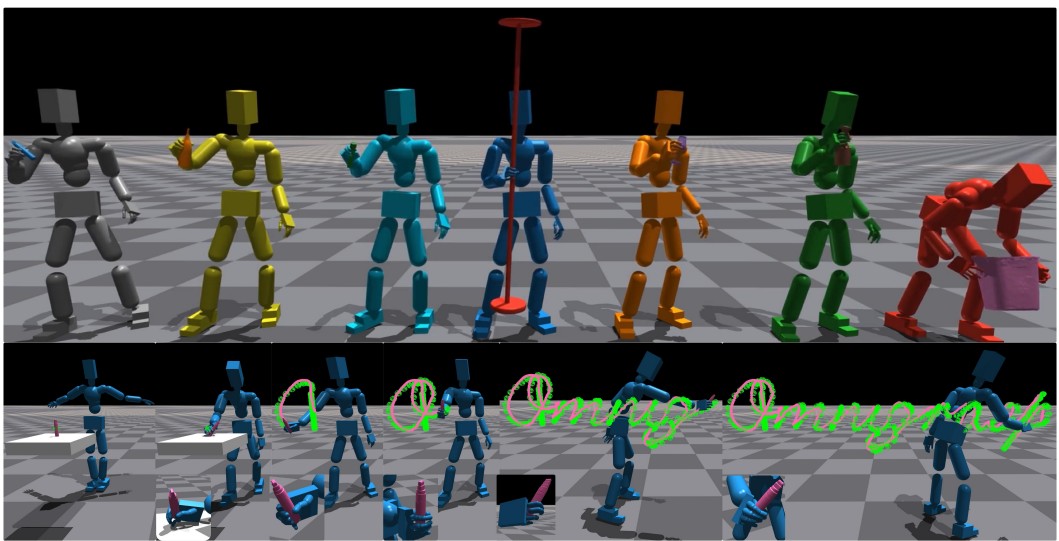

Figure 1: We control a simulated humanoid to grasp diverse objects and follow complex trajectories. (*Top*): picking up and holding objects. (*Bottom*): green dots - reference trajectory; pink dots - object trajectory.

## Abstract

We present a method for controlling a simulated humanoid to grasp an object and move it to follow an object's trajectory. Due to the challenges in controlling a humanoid with dexterous hands, prior methods often use a disembodied hand and only consider vertical lifts or short trajectories. This limited scope hampers their applicability for object manipulation required for animation and simulation. To close this gap, we learn a controller that can pick up a large number (>1200) of objects and carry them to follow randomly generated trajectories. Our key insight is to leverage a humanoid motion representation that provides human-like motor skills and significantly speeds up training. Using only simplistic reward, state, and object representations, our method shows favorable scalability on diverse objects and trajectories. For training, we do not need a dataset of paired full-body motion and object trajectories. At test time, we only require the object mesh and desired trajectories for grasping and transporting. To demonstrate the capabilities of our method, we show state-of-the-art success rates in following object trajectories and generalizing to unseen objects. Code and models will be released.

---

* Equal Contribution          † Equal Advising

38th Conference on Neural Information Processing Systems (NeurIPS 2024).

# 1 Introduction

Given an object mesh, we aim to control a simulated humanoid equipped with two dexterous hands to pick up the object and follow plausible trajectories, as shown in Fig.1. This capability could be broadly applied to creating human-object interactions for animation and AV/VR, with potential extensions to humanoid robotics [27]. However, controlling a simulated humanoid with dexterous hands for precise object manipulation poses significant challenges. The bipedal humanoid must maintain balance to enable detailed movements of the arms and fingers. Moreover, interacting with objects requires forming stable grasps that accommodate diverse object shapes. Combining these demands with the inherent difficulties of controlling a humanoid with a high degree of freedom (*e.g.* 153 DoF) significantly complicates the learning process.

These challenges have led previous methods of simulated grasping to employ a disembodied hand [16, 17, 61, 85] to grasp and transport. While this approach can generate physically plausible grasps, employing a floating hand compromises physical realism: the hands' root position and orientation are controlled by invisible forces, allowing it to remain nearly perfectly stable during grasping. Moreover, studying the hand in isolation does not accurately reflect its typical use, which is when it is attached to a mobile and flexible body. A naive approach to supporting hands is to use existing full-body motion imitators [42] to provide body control and train additional hand controllers for grasping. However, the presence of a body introduces instability, limits hand movement, and requires synchronizing the entire body to facilitate finger motion. State-of-the-art (SOTA) full-body imitators also have an average 30mm tracking error for the hands, which can cause the humanoid to miss objects. Due to the above challenges, previous work that studies full-body object manipulations often limits its scope to only one sequence of object interaction [78] and encounters difficulties in trajectory following [6], even when trained with highly specialized motion priors.

Another challenge of grasping is the diversity of the object shapes and trajectories. Each object may require a unique type of grasping, and scaling to thousands of different objects often requires training procedures such as generalist-specialist training [85] or curriculum [75, 101]. There is also infinite variability in potential object trajectories, and each trajectory may necessitate precise full-body coordination. Thus, prior work typically focuses on simple trajectories, such as vertical lifting [16, 85], or on learning a single, fixed, and pre-recorded trajectory per policy [17]. The flexibility with which humans manipulate objects to follow various trajectories while holding them remains unobtainable for current humanoids, even in simulations.

In this work, we introduce a full-body and dexterous humanoid controller capable of picking up and following diverse object trajectories using Reinforcement Learning (RL). Our proposed method, Omnigrasp, presents a scalable approach that generalizes to unseen object shapes and trajectories. Here, "Omni" refers to following any trajectory in all directions within a reasonable range and grasping diverse objects. Our key insight lies in using a pretrained universal dexterous motion representation as the action space. Directly training a policy on the joint actuation space using RL results in unnatural motion and leads to a severe exploration problem. Exploration noise in the torso can lead to a large deviation in the location of the arm and wrist as the noise propagates through the kinematic chain. This can lead to the humanoid quickly knocking the object away, which hinders training progress. Prior work has explored using a separate body and hand latent space trained using adversarial learning [6]. However, as the adversarial latent space can only cover small-scale and curated datasets, these methods do not achieve a high grasping success rate. The separation of hands and body motion prior also adds complexity to the system. We propose using a unified *universal and dexterous* humanoid motion latent space [41]. Learned from a large-scale human motion database [45], our motion representation provides a compact and efficient action space for RL exploration. We enhance the dexterity of this latent space by incorporating articulated hand motions into the existing body-only human motion dataset.

Equipped with a universal motion representation, our humanoid controller does not require any specialized interaction graph [78, 102] to learn human-object interactions. Our input to the policy consists only of object and trajectory-following information and is devoid of any grasp or reference body motion. For training, we use randomly generated trajectories and do not require paired full-body human-object motion data. We also identify the importance of pre-grasps [17] (the hand pose right before grasping) and utilize it in our reward design. The resulting policy can be directly applied to transport new objects without additional processing and achieve a SOTA success rate on following object trajectories captured by Motion Capture (MoCap).

To summarize, our contributions are: (1) we design a dexterous and universal humanoid motion representation that significantly increases sample efficiency and enables learning to grasp with simple yet effective state and reward designs; (2) we show that leveraging this motion representation, one can learn grasping policies with synthetic grasp poses and trajectories, without using any paired full-body and object motion data. (3) we demonstrate the feasibility of training a humanoid controller that can achieve a high success rate in grasping objects, following complex trajectories, scaling up to diverse training objects, and generalizing to unseen objects.

## 2 Related Works

**Simulated Humanoid Control**. Simulated humanoids can be used to create animations [26, 36, 54, 55, 56, 57, 80, 94, 102], estimate full-body pose from sensors [23, 30, 33, 40, 43, 79, 92, 93, 95], and transfer to real humanoid robots [20, 27, 28, 59, 60]. Since there are no ground truth data for joint actuation and physics simulators are often non-differentiable, model-based control [29], trajectory optimization [36, 83], and deep RL [13, 54] are used instead of supervised learning. Due to its flexibility and scalability, deep RL has been popular among efforts in simulated humanoids, where a policy/controller is trained via trial and error. Most of the previous work on humanoids does not consider articulated fingers, except for a few [3, 6, 36, 49]. A dexterous humanoid controller is essential for humanoids to perform meaningful tasks in simulation and in the real world.

**Dexterous Manipulation**. Dexterous manipulation is an essential topic in robotics [7, 8, 11, 12, 15, 16, 19, 37, 62, 75, 85, 96, 97, 98] and animation [2, 6, 34, 101]. This task usually involves pick-and-place [7, 8], lifting [75, 85, 97], articulating objects [98], and following predefined object trajectories [6, 9, 17]. Most of these efforts use a disembodied hand for grasping and employ non-physical virtual forces to control the hand. Among them, D-Grasp [16] leverages the MANO [66] hand model for physically plausible grasp synthesis and 6DoF target reaching. UniDexGrasp [85] and its followup [75] use the Shadow Hand [1]. PGDM [17] trains a grasping policy for individual object trajectories and identifies pre-grasp initialization (initializing the hand in a pose right before grasping) as a crucial factor for successful grasping. For the works that consider both hands and body, PMP [3] and PhysHOI [78] train one policy for each task or object. Braun *et al*. [6] studies a similar setting to ours but relies on MoCap human-object interaction data and only uses one hand. Compared to prior work, Omnigrasp trains one policy to transport diverse objects, supports bimanual motion, and achieves a high success rate in lifting and object trajectory following.

**Kinematic Grasp Synthesis**. Synthesizing hand grasp can be widely applied in robotics and animation. A line of work [5, 10, 10, 18, 21, 38, 47, 51, 84, 89] focuses on reconstructing and predicting grasp from images or videos, while others [52, 90] study hand grasp generation to help image generation. Among them, Manipnet and CAMS [99] predict finger poses given a hand object trajectory. TOCH [103] and GeneOH [39] denoise dynamic hand pose predictions for object interactions. More research in this area focuses on generating static or sequential hand poses with a given object as the condition [31, 70, 88]. For synthesizing body and hand poses jointly, there are limited MoCap data available [71] due to difficulties in capturing synchronized full-body and object trajectories. Some generative methods [22, 35, 69, 72, 73, 82, 91] can create paired human-object interactions, but they require initialization from the ground truth [22, 69, 82], or only predict static full-body grasps [73]. In this work, we use GrabNet [70] trained on object shapes from OakInk [86] to generate hand poses as reward guidance for our policy training.

**Humanoid Motion Representation**. Due to the high DoF of a humanoid and the sample inefficiency of RL training, the search space within which the policy operates during trial and error is crucial. A more structured action space such as motion primitives [24, 25, 48, 63] or motion latent space [56, 74] can significantly increase sample efficiency since the policy can sample coherent motion instead of relying on random "jittering" noise. This is especially important for humanoids with dexterous hands, where the torso motion can drastically affect the hand movement and lead to the humanoid knocking the object away. Thus, prior work in this space utilizes part-based motion priors [3, 6] trained on specialized datasets. While effective in the single task setting where the humanoid only needs to perform actions close to the ones in the specialized datasets, these motion priors can hardly scale to more free-formed motion, such as following randomly generated object trajectories. We extend the recently proposed universal humanoid motion representation, PULSE [41], to the dexterous humanoid setting and demonstrate that a 48-dimensional, full-body-and-hand motion latent space can be used to pick up and follow randomly generated trajectories.

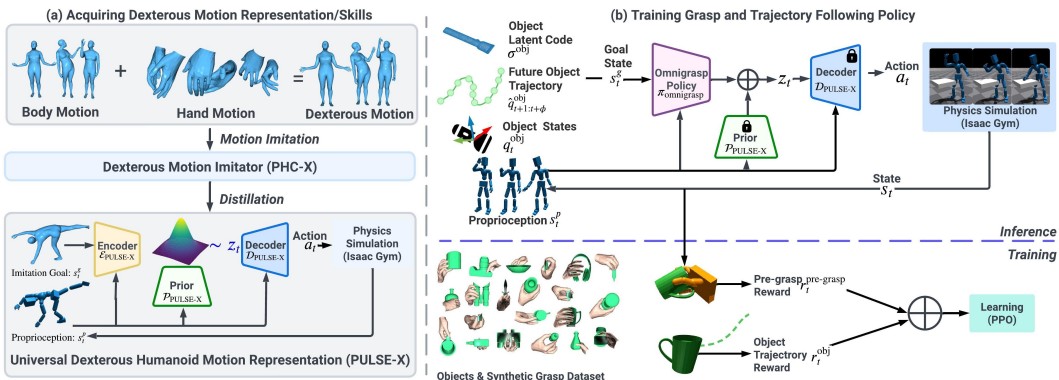

Figure 2: Omnigrasp is trained in two stages. (a) A universal and dexterous humanoid motion representation is trained via distillation. (b) Pre-grasp guided grasping training using a pretrained motion representation.

## 3 Preliminaries

We define the human pose as $\boldsymbol{q}_t \triangleq (\boldsymbol{\theta}_t, \boldsymbol{p}_t)$, consisting of 3D joint rotation $\boldsymbol{\theta}_t \in \mathbb{R}^{J \times 6}$ and position $\boldsymbol{p}_t \in \mathbb{R}^{J \times 3}$ of all $J$ links on the humanoid (hands and body), using the 6 degree-of-freedom (DOF) rotation representation [104]. To define velocities $\dot{\boldsymbol{q}}_{1:T}$, we have $\dot{\boldsymbol{q}}_t \triangleq (\boldsymbol{\omega}_t, \boldsymbol{v}_t)$ as angular $\boldsymbol{\omega}_t \in \mathbb{R}^{J \times 3}$ and linear velocities $\boldsymbol{v}_t \in \mathbb{R}^{J \times 3}$. For objects, we define their 3D trajectories $\boldsymbol{q}_t^{\text{obj}}$ using object position $\boldsymbol{p}_t^{\text{obj}}$, orientation $\boldsymbol{\theta}_t^{\text{obj}}$, linear velocity $\boldsymbol{v}_t^{\text{obj}}$, and angular velocity $\boldsymbol{\omega}_t^{\text{obj}}$. As a notation convention, we use $\hat{\cdot}$ to denote the kinematic quantities from Motion Capture (MoCap) or trajectory generator and normal symbols without accents for values from the physics simulation. $\hat{\boldsymbol{O}}$ refers to a dataset of diverse object meshes.

**Goal-conditioned Reinforcement Learning for Humanoid Control**. We define the object grasping and transporting task using the general framework of goal-conditioned RL. Namely, a goal-conditioned policy $\pi$ is trained to control a simulated humanoid to grasp an object and follow object trajectories $\hat{\boldsymbol{q}}_{1:T}^{\text{obj}}$ using dexterous hands. The learning task is formulated as a Markov Decision Process (MDP) defined by the tuple $\mathcal{M} = \langle \boldsymbol{S}, \boldsymbol{A}, \boldsymbol{T}, \mathcal{R}, \gamma \rangle$ of states, actions, transition dynamics, reward function, and discount factor. The simulation determines the state $\boldsymbol{s}_t \in \boldsymbol{S}$ and transition dynamics $\boldsymbol{T}$, where a policy computes the action $\boldsymbol{a}_t$. The state $\boldsymbol{s}_t$ contains the proprioception $\boldsymbol{s}_t^{\text{p}}$ and the goal state $\boldsymbol{s}_t^{\text{g}}$. Proprioception is defined as $\boldsymbol{s}_t^{\text{p}} \triangleq (\boldsymbol{q}_t, \dot{\boldsymbol{q}}_t, \boldsymbol{c}_t)$, which contains the 3D body pose $\boldsymbol{q}_t$, velocity $\dot{\boldsymbol{q}}_t$, and contact forces $\boldsymbol{c}_t$ on the hand. The goal state $\boldsymbol{s}_t^{\text{g}}$ is defined based on the states of the objects. When computing the states $\boldsymbol{s}_t^{\text{g}}$ and $\boldsymbol{s}_t^{\text{p}}$, all values are normalized with respect to the humanoid heading (yaw). Based on proprioception $\boldsymbol{s}_t^{\text{p}}$ and the goal state $\boldsymbol{s}_t^{\text{g}}$, we define a reward $r_t = \mathcal{R}(\boldsymbol{s}_t^{\text{p}}, \boldsymbol{s}_t^{\text{g}})$ for training the policy. We use proximal policy optimization (PPO) [68] to maximize discounted reward $\mathbb{E}\left[\sum_{t=1}^{T} \gamma^{t-1} r_t\right]$. Our humanoid follows the kinematic structure of SMPL-X [53] using the mean shape. It has 52 joints, of which 51 are actuated. 21 joints are body joints, and the remaining 30 joints are for two hands. All joints have 3 DoF, resulting in an actuation space of $\boldsymbol{a}_t \in \mathbb{R}^{51 \times 3}$. Each degree of freedom is actuated by a proportional derivative (PD) controller, and the action $\boldsymbol{a}_t$ specifies the PD target.

## 4 Omnigrasp: Grasping Diverse Objects and Follow Object Trajectories

To tackle the challenging problem of picking up objects and following diverse trajectories, we first acquire a universal dexterous humanoid motion representation in Sec.4.1. Using this motion representation, we design a hierarchical RL framework (Sec. 4.2) for grasping objects using simple[‡] state and reward designs guided by pre-grasps. Our architecture is visualized in Figure 2.

---

[‡] Here, the "simple reward" refers to not needing paired full-body-and-hand MoCap data when computing the reward, which increases complexity.

## 4.1 PULSE-X: Physics-based Universal Dexterous Humanoid Motion Representation

We introduce PULSE-X that extends PULSE [41] to the dexterous humanoid by adding articulated fingers. We first train a humanoid motion imitator [42] that can scale to a large-scale human motion dataset with finger motion. Then, we distill the motion imitator into a motion representation using a variational information bottleneck (similar to a VAE [32]).

**Data Augmentation**. Since full-body motion datasets that contain finger motion are rare (*e.g.*, 91% of the AMASS sequences do not have finger motion), we first augment existing sequences with articulated finger motion and construct a dexterous full-body motion dataset. Similarly to the process in BEDLAM [4], we randomly pair full-body motion from AMASS [45] with hand motion sampled from GRAB [71] and Re:InterHand [50] to create a dexterous AMASS dataset. Intuitively, training on this dataset increases the dexterity of the imitator and the subsequent motion representation.

**PHC-X: Humanoid Motion Imitation with Articulated Fingers**. Inspired by PHC [42], we design PHC-X $\pi_{\text{PHC-X}}$ for humanoid motion imitation with articulated fingers. For the finger joints, *we treat them similarly as the rest of the body (*e.g. *toe or wrist)* and find this formulation sufficient to acquire the dexterity needed for grasping. Formally, the goal state for training $\pi_{\text{PHC-X}}$ with RL is $s_t^{\text{g-mimic}} \triangleq (\hat{\theta}_{t+1} \ominus \theta_t, \hat{p}_{t+1} - p_t, \hat{v}_{t+1} - v_t, \hat{\omega}_{t+1} - \omega_t, \hat{\theta}_{t+1}, \hat{p}_{t+1})$, which contains the difference between proprioception and one frame reference pose $\hat{q}_{t+1}$.

**Learning Motion Representation via Online Distillation**. In PULSE [44], an encoder $\mathcal{E}_{\text{PULSE-X}}$, decoder $\mathcal{D}_{\text{PULSE-X}}$, and prior $\mathcal{P}_{\text{PULSE-X}}$ are learned to compress motor skills into a latent representation. For downstream tasks, the frozen decoder and prior will translate the latent code to joint actuation. Formally, the encoder $\mathcal{E}_{\text{PULSE-X}}(z_t|s_t^{\text{p}}, s_t^{\text{g-mimic}})$ computes the latent code distribution based on current input states. The decoder $\mathcal{D}_{\text{PULSE-X}}(a_t|s_t^{\text{p}}, z_t)$ produces action (joint actuation) based on the latent code $z_t$. The prior $\mathcal{P}_{\text{PULSE-X}}(z_t|s_t^{\text{p}})$ defines a Gaussian distribution based on proprioception and replaces the unit Gaussian distribution used in VAEs [32]. The prior increases the expressiveness of the latent space and guides downstream task learning by forming a residual action space (see Sec.4.2). We model the encoder and prior distribution as diagonal Gaussian:

$$\mathcal{E}_{\text{PULSE-X}}(z_t|s_t^{\text{p}}, s_t^{\text{g-mimic}}) = \mathcal{N}(z_t|\mu_t^e, \sigma_t^e), \mathcal{P}_{\text{PULSE-X}}(z_t|s_t^{\text{p}}) = \mathcal{N}(z_t|\mu_t^p, \sigma_t^p). \tag{1}$$

To train the models, we use online distillation similar to DAgger [67] by rolling out the encoder-decoder in simulation and querying $\pi_{\text{PHC-X}}$ for action labels $a_t^{\text{PHC-X}}$. For more information and evaluation of PHC-X and PULSE-X, please refer to the Appendix B.

## 4.2 Pre-grasp Guided Object Manipulation

Using hierarchical RL and PULSE-X's trained decoder $\mathcal{D}_{\text{PULSE-X}}$ and prior $\mathcal{P}_{\text{PULSE-X}}$, the action space for our object manipulation policy becomes the latent motion representation $z_t$. Since the action space serves as a strong human-like motion prior, we can use simple state and reward design and do not require any paired object and human motion to learn grasping policies. We use only hand pose before grasping (pregraps), either from a generative method or MoCap, to train our policy.

**State**. To provide the task policy $\pi_{\text{Omnigrasp}}$ with information about the object and the desired object trajectory, we define the goal state as

$$s_t^{\text{g}} \triangleq (\hat{p}_{t+1:t+\phi}^{\text{obj}} - p_t^{\text{obj}}, \hat{\theta}_{t+1:t+\phi}^{\text{obj}} \ominus \theta_t^{\text{obj}}, \hat{v}_{t+1:t+\phi}^{\text{obj}} - v_t^{\text{obj}}, \hat{\omega}_{t+1:t+\phi}^{\text{obj}} - \omega_t^{\text{obj}}, p_t^{\text{obj}}, \theta_t^{\text{obj}}, \sigma^{\text{obj}}, p_t^{\text{obj}} - p_t^{\text{hand}}), \tag{2}$$

which contains the reference object pose and the difference between the reference object trajectory for the next $\phi$ frames and the current object state. $\sigma^{\text{obj}} \in \mathcal{R}^{512}$ is the object shape latent code computed using the canonical object pose and Basis Point Set (BPS) [58]. $p_t^{\text{obj}} - p_t^{\text{hand}}$ is the difference between the current object position and each hand joint position. All values are normalized with respect to the humanoid heading. Notice that the state $s_t^{\text{g}}$ does not contain body pose, grasp, or phase variables [6], which makes our method applicable to unseen objects and reference trajectories at test time.

**Action**. Similar to downstream task policies in PULSE, we form the action space of $\pi_{\text{Omnigrasp}}$ as the residual action with respect to prior's mean $\mu_t^p$ and compute the PD target $a_t$:

$$a_t = \mathcal{D}_{\text{PULSE-X}}\big(\pi_{\text{Omnigrasp}}(z_t^{\text{omnigrasp}}|s_t^{\text{p}}, s_t^{\text{g}}) + \mu_t^p\big), \tag{3}$$

**Algo 1:** Learn Omnigrasp

```
1  Function TrainOmnigrasp(𝒟_PULSE-X, 𝒫_PULSE-X, π_Omnigrasp, Ô, 𝒯^3D):
2      Input: Pretrained PULSE-X's decoder 𝒟_PULSE-X and prior 𝒫_PULSE-X, Object mesh dataset Ô, 3D trajectory Generator 𝒯^3D ;
3      while not converged do
4          M ← ∅ initialize sampling memory ;
5          while M not full do
6              q₀^obj, p̂^pre-grasp, sₜ^p ∼ randomly sample initial object state, pre-grasp, and humanoid state ;
7              q̂₁:T^obj ∼ sample reference object trajectory using 𝒯^3D ;
8              for t ← 1…T do
9                  zₜ^omnigrasp ∼ π_Omnigrasp(zₜ^omnigrasp|sₜ^p, sₜ^g)    // use pretrained latent space as action space;
10                 μₜ^p, σₜ^p ← 𝒫_PULSE-X(zₜ|sₜ^p)                       // compute prior latent code;
11                 aₜ ← 𝒟_PULSE-X(aₜ|sₜ^p, zₜ^omnigrasp + μₜ^p)         // decode action using pretrained decoder;
12                 sₜ₊₁ ← 𝒯(sₜ₊₁|sₜ, aₜ)                               // simulation;
13                 rₜ ← ℛ(sₜ^p, sₜ^g)                                  // compute reward;
14                 store (sₜ, zₜ^omnigrasp, rₜ, sₜ₊₁) into memory M ;

15         π_Omnigrasp ← PPO update using experiences collected in M ;
16         Ô_hard ← Eval and pick hard object subset to train on.

17     return π_Omnigrasp ;
```

where $\boldsymbol{\mu}_t^p$ is computed by the prior $\mathcal{P}_{\text{PULSE-X}}(\boldsymbol{z}_t|\boldsymbol{s}_t^{\text{p}})$. The policy $\pi_{\text{Omnigrasp}}$ computes $\boldsymbol{z}_t^{\text{omnigrasp}} \in \mathcal{R}^{48}$ instead of the target $\boldsymbol{a}_t \in \mathcal{R}^{51\times3}$ directly, and leverages the latent motion representation of PULSE-X to produce human-like motion.

**Reward**. While our policy does not take any grasp guidance or reference body trajectory *as input*, we utilize pre-grasp guidance in the *reward*. We refer to pre-grasp $\hat{\boldsymbol{q}}^{\text{pre-grasp}} \triangleq (\hat{\boldsymbol{p}}^{\text{pre-grasp}}, \hat{\boldsymbol{\theta}}^{\text{pre-grasp}})$ as a single frame of hand pose consisting of hand translation $\hat{\boldsymbol{p}}^{\text{pre-grasp}}$ and rotation $\hat{\boldsymbol{\theta}}^{\text{pre-grasp}}$. PGDM [17] shows that initializing a floating hand to pre-grasps can help the policy better reach objects and initiate manipulation. As we do not initialize the humanoid with the pre-grasp pose as in PGDM, we design a stepwise pre-grasp reward:

$$\boldsymbol{r}_t^{\text{omnigrasp}} = \begin{cases} r_t^{\text{approach}}, & \|\hat{\boldsymbol{p}}^{\text{pre-grasp}} - \boldsymbol{p}_t^{\text{hand}}\|_2 > 0.2 \text{ and } t < \lambda \\ r_t^{\text{pre-grasp}}, & \|\hat{\boldsymbol{p}}^{\text{pre-grasp}} - \boldsymbol{p}_t^{\text{hand}}\|_2 \le 0.2 \text{ and } t < \lambda \\ r_t^{\text{obj}}, & t \ge \lambda, \end{cases} \tag{4}$$

based on time and the distance between the object and hands. Here, $\lambda = 1.5s$ indicates the frame in which grasping should occur, and $\boldsymbol{p}_t^{\text{hand}}$ indicates the hand position. When the object is far away from the hands ($\|\hat{\boldsymbol{p}}^{\text{pre-grasp}} - \boldsymbol{p}_t^{\text{hand}}\|_2 > 0.2$), we use an approach reward $r_t^{\text{approach}}$ similar to a point-goal [42, 81] reward $r_t^{\text{approach}} = \|\hat{\boldsymbol{p}}^{\text{pre-grasp}} - \boldsymbol{p}_t^{\text{hand}}\|_2 - \|\hat{\boldsymbol{p}}^{\text{pre-grasp}} - \boldsymbol{p}_{t-1}^{\text{hand}}\|_2,$, where the policy is encouraged to get close to the pre-grasp. After the hands are close enough ($\le 0.2$m), we use a more precise hand imitation reward: $r_t^{\text{pre-grasp}} = w_{\text{hp}}e^{-100\|\hat{\boldsymbol{p}}^{\text{pre-grasp}} - \boldsymbol{p}_t^{\text{hand}}\|_2 \times \mathbb{1}\{\|\hat{\boldsymbol{p}}^{\text{pre-grasp}} - \hat{\boldsymbol{p}}_t^{\text{obj}}\|_2 \le 0.2\}} + w_{\text{hr}}e^{-100\|\boldsymbol{\theta}^{\text{pre-grasp}} - \boldsymbol{\theta}_t^{\text{hand}}\|_2}$, to encourage the hands to be close to pre-grasps. For grasps that involve only one hand, we use an indicator variable $\mathbb{1}\{\|\hat{\boldsymbol{p}}^{\text{pre-grasp}} - \hat{\boldsymbol{p}}_t^{\text{obj}}\|_2 \le 0.2\}$ to filter out hands that are too far away from the object. After timestep $\lambda$, we use only the object trajectory following reward:

$$r_t^{\text{obj}} = (w_{\text{op}}e^{-100\|\hat{\boldsymbol{p}}_t^{\text{obj}} - \boldsymbol{p}_t^{\text{obj}}\|_2} + w_{\text{or}}e^{-100\|\hat{\boldsymbol{\theta}}_t^{\text{obj}} - \boldsymbol{\theta}_t^{\text{obj}}\|_2} + w_{\text{ov}}e^{-5\|\hat{\boldsymbol{v}}_t^{\text{obj}} - \boldsymbol{v}_t^{\text{obj}}\|_2} + w_{\text{oav}}e^{-5\|\hat{\boldsymbol{\omega}}_t^{\text{obj}} - \boldsymbol{\omega}_t^{\text{obj}}\|_2}) \cdot \mathbb{1}\{\text{C}\} + \mathbb{1}\{\text{C}\} \cdot w_{\text{c}}. \tag{5}$$

$r_t^{\text{obj}}$ computes the difference between the current and reference object pose, which is filtered by an indicator variable $\mathbb{1}\{\text{C}\}$ that is set to true if the object is in contact with the humanoid hands. The reward $\mathbb{1}\{\text{C}\} \cdot w_{\text{c}}$ encourages the humanoid's hand to have contact with the object. Hyperparameters can be found in Appendix C.

**Object 3D Trajectory Generator**. As there is a limited number of ground-truth object trajectories [17], either collected from MoCap or animators, we design a 3D object trajectory generator that can create trajectories with varying speed and direction. Using the trajectory generator, our policy can be trained without any ground-truth object trajectories. This strategy provides better coverage of potential object trajectories, and the resulting policy achieves higher success in following unseen trajectories (see Table 1). Specifically, we extend the 2D trajectory generator used in PACER [65, 76] to 3D, and create our trajectory generator $\mathcal{T}^{\text{3D}}(\boldsymbol{q}_0^{\text{obj}}) = \hat{\boldsymbol{q}}_{1:T}^{\text{obj}}$. Given initial object pose $\boldsymbol{q}_0^{\text{obj}}$, $\mathcal{T}^{\text{3D}}$ can generate a sequence of plausible reference object motion $\hat{\boldsymbol{q}}_{1:T}^{\text{obj}}$. We limit the z-direction trajectory to between 0.03m and 1.8m and leave the xy direction unbounded. For more information and sampled trajectories, please refer to Appendix C.

Table 1: Quantitative results on object grasp and trajectory following on the GRAB dataset.

| Method | Traj | GRAB-Goal-Test (Cross-Object, 140 sequences, 5 unseen objects) | | | | | | | GRAB-IMoS-Test (Cross-Subject, 92 sequences, 44 objects) | | | | | | |
|---|---|---|---|---|---|---|---|---|---|---|---|---|---|---|---|
| | | $Succ_{grasp}\uparrow$ | $Succ_{traj}\uparrow$ | TTR↑ | $E_{pos}\downarrow$ | $E_{rot}\downarrow$ | $E_{acc}\downarrow$ | $E_{vel}\downarrow$ | $Succ_{grasp}\uparrow$ | $Succ_{traj}\uparrow$ | TTR↑ | $E_{pos}\downarrow$ | $E_{rot}\downarrow$ | $E_{acc}\downarrow$ | $E_{vel}\downarrow$ |
| PPO-10B | Gen | 98.4% | 55.9% | 97.5% | 36.4 | **0.4** | 21.0 | 14.5 | 96.8% | 53.2% | 97.9% | 35.6 | **0.5** | 19.6 | 13.9 |
| PHC [42] | MoCap | 3.6% | 11.4% | 81.1% | 66.3 | 0.8 | 1.5 | 3.8 | 0% | 3.3% | 97.4% | 56.5 | 0.3 | 1.4 | 2.9 |
| AMP [57] | Gen | 90.4% | 46.6% | 94.0% | 40.7 | 0.6 | 5.3 | 5.3 | 95.8% | 49.2% | 96.5% | 34.9 | 0.5 | 6.2 | 6.0 |
| Braun et al. [6] | MoCap | 79% | - | 85% | - | - | - | - | 64% | - | 65% | - | - | - | - |
| Omnigrasp | MoCap | 94.6% | 84.8% | 98.7% | **28.0** | **0.5** | **4.2** | **4.3** | 95.8% | 85.4% | 99.8% | **27.5** | **0.6** | **5.0** | **5.0** |
| Omnigrasp | Gen | **100%** | **94.1%** | **99.6%** | 30.2 | 0.93 | 5.4 | 4.7 | **98.9%** | **90.5%** | 99.8% | 27.9 | 0.97 | 6.3 | 5.4 |

**Training**. Our training process is depicted in Algo 1. One of the main sources of performance improvement for motion imitation is hard-negative mining [42, 43], where the policy is evaluated regularly to find the failure sequences to train on. Thus, instead of using object curriculum [75, 85, 101], we use a simple hard-negative mining process to pick hard objects $\hat{O}_{hard}$ to train on. Specifically, let $s_j$ be the number of failed lifts for object $j$ over all previous runs. The probability of choosing object $j$ among all objects is $P(j) = \frac{s_j}{\sum_i^J s_i}$.

**Object and Humanoid Initial State Randomization**. Since objects can have diverse initial positions and orientations with respect to the humanoid, it is crucial to have the policy exposed to diverse initial object states. Given the object dataset $\hat{O}$ and the provided initial states (either from MoCap or by dropping the object in simulation) $q_0^{obj}$, we perturb $q_0^{obj}$ by adding randomly sampled yaw-direction rotation and adjusting the position component $q_0^{obj}$. We do not change the pitch and yaw of the object's initial pose as some poses are invalid in simulation. For the humanoid, we use the initial state from the dataset if provided (*e.g.* GRAB dataset [71]), and a standing T-pose if there is no paired data.

**Inference**. During inference, the object latent code $p_t^{obj}$, a random object starting pose $q_0^{obj}$, and desired object trajectory $\hat{q}_{1:T}^{obj}$ is all that is required, without any dependency on pre-grasps or paired kinematic human pose.

# 5   Experiments

**Datasets**. We use the GRAB [71], OakInk [86], and OMOMO [34] to study grasping small and large objects. The GRAB dataset contains 1.3k paired full-body motion and object trajectories of 50 objects (we remove the doorknob as it is not movable). Since the GRAB dataset provides reference body and object motion, we use them to extract initial humanoid positions and pre-grasps. We follow prior art [6] in constructing cross-object (45 for training and 5 for testing) and cross-subject (9 subjects for training and 1 for testing) train-test sets. On GRAB, we evaluate on following MoCap object trajectories using the mean body shape humanoid. The OakInk dataset contains 1700 diverse objects of 32 categories with real-world scanned and generated object meshes. We split them into 1330 objects for training, 185 for validation, and 185 for testing. Train-test splits are conducted within categories, with train and test splits containing objects from all categories. Since no paired MoCap human motion or grasps exists for the OakInk dataset, we use an off-the-shelf grasp generator [86] to create pre-grasps. The OMOMO dataset contains 15 large objects (table lamps, monitors, *etc.*) with reconstructed mesh, and we pick 7 of them that have cleaner meshes. Due to the limited number of objects from OMOMO, we only test lifting on the objects used for training to verify that our pipeline can learn to move larger objects. On OMOMO and OakInk, we study vertical lifting (30cm) and holding (3s) as the trajectory for quantitative results.

**Implementation Details**. Simulation is conducted in Isaac Gym [46], where the policy is run at 30 Hz and the simulation at 60 Hz. For PULSE-X and PHC-X, each policy is a 6-layer MLP. For the grasping task, we employ a GRU [14] based recurrent policy and use a GRU with a latent dimension of 512, followed by a 3-layer MLP. We train Omnigrasp for three days collecting around $10^9$ samples on a Nvidia A100 GPU. PHC-X and PULSE-X are trained once and frozen, which takes around 1.5 weeks and 3 days. Object density is 1000 kg/m$^3$. The static and dynamic friction coefficients of the object and humanoid fingers are set to 1. For reference object trajectory, we use $\phi = 20$ future frames sampled at 15Hz. For more details, please refer to Appendix C.

**Metrics**. For the object trajectory following, we report the position error $E_{pos}$ (mm), rotation error $E_{rot}$ (radian), and physics-based metrics such as acceleration error $E_{acc}$ (mm/frame$^2$) and velocity error $E_{vel}$ (mm/frame). Following prior art in full-body simulated humanoid grasping [6], we report the grasp success rate $Succ_{grasp}$ and Trajectory Targets Reached (TTR). The grasp success rate $Succ_{grasp}$ deems a grasp successful when the object is held for at least 0.5s in the physics simulation without dropping. TTR measures the ratio of the target position (< 12cm away from the target position) reached over all the time steps in the trajectory and is only measured on successful trajectories. To measure the complete trajectory success rate, we also report $Succ_{traj}$, where a trajectory following is unsuccessful if, at any point in time, the object is > 25cm away from the reference.

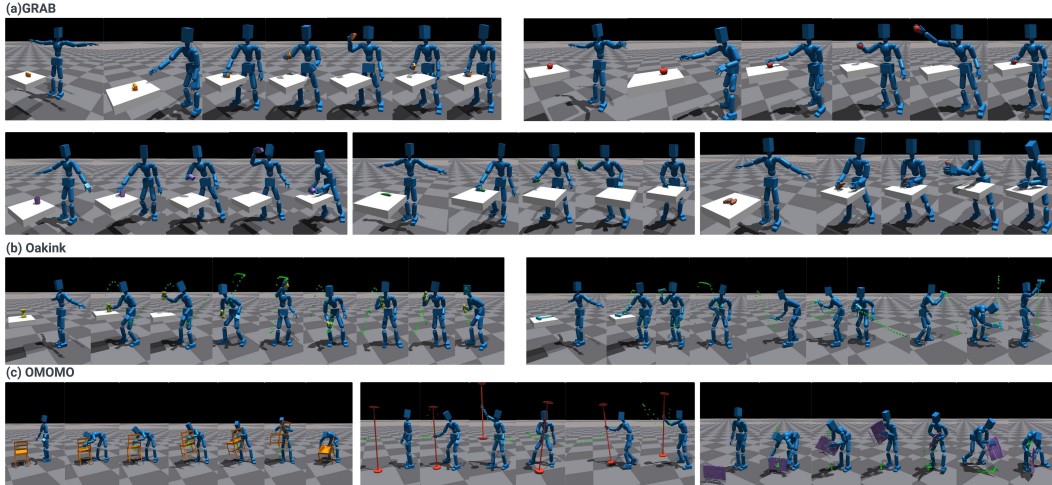

Figure 3: Qualitative results. Unseen objects are tested for GRAB and OakInk. Green dots: reference trajectories. Best seen in videos on our `supplement site`.

## 5.1 Grasping and Trajectory Following

As motion is best seen in videos, please refer to `supplement site` for extended evaluation on trajectory following, unseen objects, and robustness. Unless otherwise specified, all policies are trained on their respective dataset training split, and we conduct cross-dataset experiments on GRAB and OakInk. All experiments are run 10 times and averaged as the simulator yields slightly different results for each run due to *e.g.* floating-point error. As full-body simulated humanoid grasping is a relatively new task with a limited number of baselines, we use Braun et [6] as our main comparison. We also implement AMP [57] and PHC [42] as baselines. We train AMP with a similar state and reward design (without using PULSE-X's latent space) and a task and discriminator reward weighting of 0.5 and 0.5. PHC refers to using an imitator for grasping, where we directly feed ground-truth kinematic body and finger motion to a pretrained imitator to grasp objects. Since PHC and PULSE-X require pre-training, we also include PPO-10B, which is trained using RL without PULSE-X for a month (∼10 billion samples).

**GRAB Dataset (50 objects)**. Since Braun *et al*. do not use randomly generated trajectories, we train Omnigrasp using two different settings for a fair comparison: one trained with MoCap object trajectories only, and one trained using synthetic trajectories only. From Table 1, we can see that our method outperforms prior SOTA and baselines on all metrics, especially on success rate and trajectory following. Since all methods are simulation-based, we omit penetration/foot sliding metrics and report the precise trajectory tracking errors instead. Training directly using PPO without PULSE-X leads to a performance that significantly lags behind Omnigrasp, even though it has used similar aggregate samples (counting PHC-X and PULSE-X training). Compared to Braun *et al*., Omnigrasp achieves a high success rate on both object lifting and trajectory following. Directly using the motion imitator, PHC, yields a low success rate even when the ground-truth kinematic pose is provided, showing that the imitator's error (on average 30mm) is too large to overcome for precise object grasping. The body shape mismatch between MoCap and our simulated humanoid also contributes to this error. AMP leads to a low trajectory success rate, showing the importance of using a motion prior in the *actions space*. Omnigrasp can track the MoCap trajectory precisely with an average error of 28mm. Comparing training on MoCap trajectories and randomly generated ones, we can see that training on generated trajectories achieves better performance on success rate and position error, though worse on rotation error. This is due to our 3D trajectory generator offering good coverage on physically plausible 3D trajectories, but there is a gap between the randomly generated rotations and MoCap object rotation. This can be improved by introducing more rotation variation on the trajectory generator. The gap between trajectory $Succ_{traj}$ and grasp success $Succ_{grasp}$ shows that following the full trajectory is a much harder task than just grasping, and the object can be dropped during trajectory following. Qualitative results can be found in Fig. 3.

**OakInk Dataset (1700 objects)**. On the OakInk dataset, we scale our grasping policy to >1000 objects and test our generalization to unseen objects. We also conduct cross-dataset experiments, where we train on the GRAB dataset and test on the OakInk dataset. Results are shown in Table 3. We can see that 1272 out of the 1330 objects are trained to be picked up, and the whole lifting process also has a high success rate. We observe similar results on the test split. Upon inspection, the failed objects are usually either too large or too small for the humanoid to establish a grasp. The large number of objects also places a strain on the hard-negative mining process. The policy trained on both GRAB and OakInk shows the highest success rate, as on GRAB, there are bi-manual pre-grasps, and the policy learned to use both hands.

Table 3: Quantitative results on OakInk with our method. We also test Omnigrasp cross-dataset, where a policy trained on GRAB is tested on the OakInk dataset.

| Training Data | OakInk-Train (1330 objects) | | | | | | | OakInk-Test (185 objects) | | | | | | |
|---|---|---|---|---|---|---|---|---|---|---|---|---|---|---|
| | $Succ_{grasp}$ ↑ | $Succ_{traj}$ ↑ | TTR ↑ | $E_{pos}$ ↓ | $E_{rot}$ ↓ | $E_{acc}$ ↓ | $E_{vel}$ ↓ | $Succ_{grasp}$ ↑ | $Succ_{traj}$ ↑ | TTR ↑ | $E_{pos}$ ↓ | $E_{rot}$ ↓ | $E_{acc}$ ↓ | $E_{vel}$ ↓ |
| OakInk | 93.7% | 86.2% | **100%** | 21.3 | **0.4** | 7.7 | 6.0 | **94.3%** | 87.5% | **100%** | 21.2 | **0.4** | 7.6 | 5.9 |
| GRAB | 84.5% | 75.2% | 99.9% | 22.4 | **0.4** | 6.8 | 5.7 | 81.9% | 72.1% | 99.9% | 22.7 | **0.4** | 7.1 | 5.8 |
| GRAB + OakInk | **95.6%** | **92.0%** | **100%** | **21.0** | 0.6 | **5.4** | **4.8** | 93.5% | **89.0%** | **100%** | 21.3 | 0.6 | **5.4** | **4.8** |

Table 4: Ablation on various strategies of training Omnigrasp. PULSE-X: whether to use the latent motion representation. pre-grasp: pre-grasp guidance reward. Dex-AMASS: whether to train PULSE-X on the dexterous AMASS dataset. Rand-pose: randomizing the object initial pose. Hard-neg: hard-negative mining.

| | GRAB-Goal-Test (Cross-Object, 140 sequences, 5 unseen objects) | | | | | | | | | | |
|---|---|---|---|---|---|---|---|---|---|---|---|
| idx | PULSE-X | pre-grasp | Dex-AMASS | Rand-pose | Hard-neg | $Succ_{grasp}$ ↑ | $Succ_{traj}$ ↑ | TTR ↑ | $E_{pos}$ ↓ | $E_{rot}$ ↓ | $E_{acc}$ ↓ | $E_{vel}$ ↓ |
| 1 | ✗ | ✓ | ✓ | ✓ | ✓ | 97.0% | 33.6% | 92.8% | 43.5 | **0.5** | 10.6 | 8.3 |
| 2 | ✓ | ✗ | ✓ | ✓ | ✓ | 77.1% | 57.9% | 97.4% | 54.9 | 1.0 | 5.5 | 5.2 |
| 3 | ✓ | ✓ | ✗ | ✓ | ✓ | 94.4% | 77.3% | 99.3% | 30.5 | 0.9 | 4.8 | **4.4** |
| 4 | ✓ | ✓ | ✓ | ✗ | ✓ | 92.9% | 79.9% | 99.2% | 31.4 | 1.1 | **4.5** | **4.4** |
| 5 | ✓ | ✓ | ✓ | ✓ | ✗ | 94.0% | 71.6% | 98.4% | 32.3 | 1.3 | 6.2 | 5.7 |
| 6 | ✓ | ✓ | ✓ | ✓ | ✓ | **100%** | **94.1%** | **99.6%** | **30.2** | 0.9 | 5.4 | 4.7 |

Using both hands significantly improves the success rate on some larger objects, where the humanoid can scoop up the object with one hand and carry it with both. As OakInk only has pre-grasps using one hand, it cannot learn such a strategy. Surprisingly, training on only GRAB achieves a high success rate on OakInk, picking up more than 1000 objects without training on the dataset, showcasing the robustness of our grasping policy on unseen objects.

Table 2: Quantitative results on the OMOMO dataset.

| OMOMO (7 objects) | | | | | | |
|---|---|---|---|---|---|---|
| $Succ_{grasp}$ ↑ | $Succ_{traj}$ ↑ | TTR ↑ | $E_{pos}$ ↓ | $E_{rot}$ ↓ | $E_{acc}$ ↓ | $E_{vel}$ ↓ |
| 7/7 | 7/7 | 100% | 22.8 | 0.2 | 3.1 | 3.3 |

**OMOMO Dataset (7 objects)**. On the OMOMO dataset, we train a policy to show that our method can learn to pick up large objects. Table 2 shows that our method can successfully learn to pick up all the objects, including chairs and lamps. For larger objects, the pre-grasp guidance is essential for guiding the policy to learn bi-manual manipulation skills (as is shown in Fig 3)

## 5.2 Ablation and Analysis

**Ablation**. In this section, we study the effects of different components of our framework using the cross-object split of the GRAB dataset. Results are shown in Table 4. First, we compare our method trained with (Row 6) or without (R1) PULSE-X's action space. Using the same reward and state design, we can see that using the universal motion prior significantly improves success rates. Upon inspection, using PULSE-X also yields human-like motion, while not using it leads to unnatural motion (see in `supplement site`). R2 vs. R6 shows that the pre-grasp guidance is essential in learning grasps that are stable for grasping objects, but without it, some objects can still be grasped successfully. The difference between R3 and R6 is whether to train using the dexterous AMASS dataset. R3 vs R6 shows that without training on a dataset that has diverse hand motion and full-body motion, the policy can learn to pick up objects (high grasp success rate), but struggles in trajectory following. This is expected as the motion prior probably lacks the motion of "holding the object while moving". R4 and R5 show that object position randomization and hard-negativing mining are crucial for learning robust and successful policies. Ablations on the object latent code, RNN policy, *etc*. can be found in the Appendix C.

**Analysis: Diverse Grasps**. In Fig. 4, we visualize the grasping strategy used by our method. We can see that based on the object shape, our policy uses a diverse set of grasping strategies to hold the object during the trajectory following. Based on the trajectory and object initial pose, Omnigrasp discovers different grasping poses for the *same* object, showcasing the advantage of using simulation and laws of physics for grasp generation. We also notice that for larger objects, our policy will resort to using two hands and a non-prehensile transport strategy. This behavior is learned from pre-grasps in GRAB, which utilize both hands for object manipulation.

**Analysis: Robustness and Potential for Sim-to-real Transfer**. In Table 5, we add uniform random noise [-0.01, 0.01] to both task observation (positions, object latent codes, etc.) and proprioception. A similar scale (0.01) of random noise is used in sim-to-real RL to tackle noisy input in real-world humanoids [28]. We see that Omnigrasp is relatively robust to input noise, even though it has not been trained with noisy input. Performance drop is more prominent in the acceleration and velocity metrics. Adding noise during training can further improve robustness. We do not claim that Omnigrasp is currently ready for real-world deployment, but we

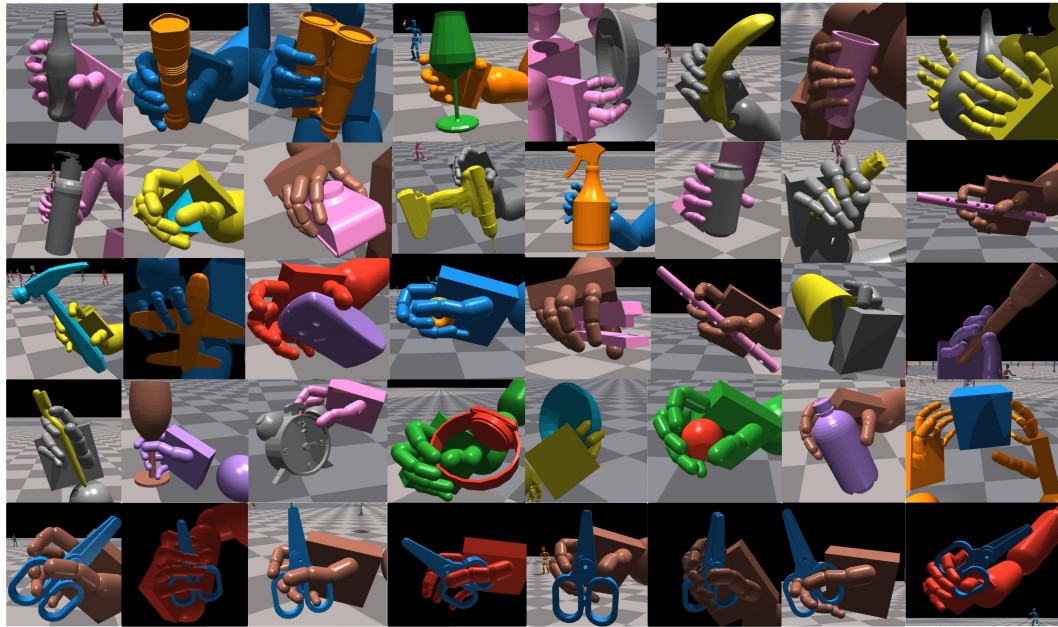

Figure 4: *(Top rows)*: grasping different objects using both hands. *(Bottom)* diverse grasps on the same object.

Table 5: Study on how noise affects pretrained Omnigrasp Policy

| Method | Noise Scale | GRAB-Goal-Test (Cross-Object, 140 sequences, 5 unseen objects) | | | | | | | GRAB-IMoS-Test (Cross-Subject, 92 sequences, 44 objects) | | | | | | |
| | | $Succ_{grasp}$ ↑ | $Succ_{traj}$ ↑ | TTR ↑ | $E_{pos}$ ↓ | $E_{rot}$ ↓ | $E_{acc}$ ↓ | $E_{vel}$ ↓ | $Succ_{grasp}$ ↑ | $Succ_{traj}$ ↑ | TTR ↑ | $E_{pos}$ ↓ | $E_{rot}$ ↓ | $E_{acc}$ ↓ | $E_{vel}$ ↓ |
| Omnigrasp | 0 | **100%** | **94.1%** | **99.6%** | **30.2** | **0.93** | **5.4** | **4.7** | 98.9% | **90.5%** | **99.8%** | **27.9** | **0.97** | **6.3** | **5.4** |
| Omnigrasp | 0.01 | **100%** | 91.4% | 99.2% | 34.8 | 1.1 | 15.6 | 11.5 | **99.5%** | 86.2% | 99.6% | 32.5 | 1.0 | 17.9 | 13.2 |

believe that a similar system design plus sim-to-real modifications (e.g. domain randomization, distilling into a vision-based policy) has the potential. We conduct more analysis on the robustness of our method with respect to initial object position, object weight, and object trajectories on our `supplement site`.

# 6 Limitations, Conclusions, and Future Work

**Limitations**. While Omnigrasp demonstrates the feasibility of controlling a simulated humanoid to grasp diverse objects and hold them to follow diverse trajectories, many limitations remain. For example, though the 6DoF input is provided in the input and reward, the rotation error remains to be further improved. Omnigrasp has yet to support precise in-hand manipulations. The success rate on trajectory following can be improved, as objects can be dropped or not picked up. Another area of improvement is to achieve *specific* types of grasps on the object, which may require additional input such as desired contact points and grasp. Human-level dexterity, even in simulation, remains challenging. For visualization of failure cases, see `supplement site`.

**Conclusion and Future Work**. In conclusion, we present Omnigrasp, a humanoid controller capable of grasping > 1200 objects and following trajectories while holding the object. It generalizes to unseen objects of similar sizes, utilizes bi-manual skills, and supports picking up larger objects. We demonstrate that by using a pretrained universal humanoid motion representation, grasping can be learned using simplistic reward and state designs. Future work includes improving trajectory following success rate, improving grasping diversity, and supporting more object categories. Also, improving upon the humanoid motion representation is a promising direction. While we utilize a simple yet effective unified motion latent space, separating the motion representation for hands and body [3, 6] could lead to further improvements. Effective object representation is also an important future direction. How to formulate an object representation that does not rely on canonical object pose and generalizes to vision-based systems will be valuable to help the model generalize to more objects.

**Acknowledgement**. Zhengyi Luo is supported by the Meta AI Mentorship (AIM) program.

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

# Appendix

## A   Introduction

In this document, we include additional details about Omnigrasp that are omitted from the main paper due to the page limit. In Sec.B, we include additional information about training and evaluating the performance of our humanoid motion representation, PULSE-X. In Sec. C, we include details about Omnigrasp, such as the trajectory generator and training procedures.

Extensive qualitative results are provided at the project page as well as the supplementary zip files (which contain lower-resolution videos due to file size limitations). As motion is best seen in videos, we highly encourage our readers to view them to judge the capabilities of our method better. Specifically, we visualize using our controller to trace the characters "Omnigrasp" in the air while holding unseen objects during training. This complex trajectory is never seen during training. We also visualize the policy on GRAB [71], OakInk [87], and OMOMO [34] datasets, both for training and testing objects. On the GRAB dataset, we follow MoCap trajectories, while for the OakInk and OMOMO datasets, we showcase randomly generated trajectories for training. To demonstrate robustness to different object poses, weights, and directions, we also test our method by varying these variables and show that it can still pick up objects. Interestingly, we notice that our method prefers to use both hands to pick and hold the object as the weight of the object increases. We also include motion imitation and random motion sampling for PHC-X and PULSE-X. Further, we visualize our constructed dexterous AMASS dataset and the motion imitation result. Last, we include failure cases for grasping and trajectory following.

## B   Details about PHC-X and PULSE-X

**Data Cleaning**. To train both PHC-X and PULSE-X, we follow PULSE's [41] procedure in filtering on implausible motion. This process yields 14889 motion sequences from the AMASS dataset for training our humanoid motion representation. Out of all 14889 sequences, only 9% of the sequences contain hand motion, and training on it will bias the motion imitator to have limited dexterity. Thus, we construct the dexterous AMASS dataset by pairing hand-only motion with body-only motion and demonstrate its effectiveness in learning a motion representation that enables object grasping.

### B.1   Training and Architecture

The state, action, and rewards for PHC-X and PULSE-X follow the implementation choices of PULSE with the only modifications on the training data (dexterous AMASS) and humanoid (SMPL-X). PHC-X is trained for 1.5 week while PULSE-X takes 3 days. We use the same-sized networks: 6-layer MLP of units [2048, 1536, 1024, 1024, 512, 512] for PHC-X and 3-layer MLP of units [3096, 2048, 1024] for PULSE-X's encoder and decoders. We notice that due to the increase in DoF from SMPL (69) to SMPL-X (153), simulation is $\sim$2 times slower.

Table 7: Hyperparameters for Omnigrasp, PHC-X, and PULSE-X. $\sigma$: fixed variance for policy. $\gamma$: discount factor. $\epsilon$: clip range for PPO.

| Method | Batch Size | Learning Rate | $\sigma$ | $\gamma$ | $\epsilon$ | | | | | | # of samples |
|---|---|---|---|---|---|---|---|---|---|---|---|
| PHC-X | 3072 | $2 \times 10^{-5}$ | 0.05 | 0.99 | 0.2 | | | | | | $\sim 10^{10}$ |
| | Batch Size | Learning Rate | Latent size | | | | | | | | # of samples |
| PULSE-X | 3072 | $5 \times 10^{-4}$ | 48 | | | | | | | | $\sim 10^{9}$ |
| | Batch Size | Learning Rate | $\sigma$ | $\gamma$ | $\epsilon$ | $w_{\mathrm{op}}$ | $w_{\mathrm{or}}$ | $w_{\mathrm{ov}}$ | $w_{\mathrm{oav}}$ | $w_{\mathrm{c}}$ | # of samples |
| Omnigrasp | 3072 | $5 \times 10^{-4}$ | 0.05 | 0.99 | 0.2 | 0.5 | 0.3 | 0.05 | 0.05 | 0.1 | $\sim 10^{9}$ |

## B.2 Evaluation

We evaluate PULSE-X and PHC-X on our constructed dexterous AMASS dataset. The metrics we use are the mean per-joint position error (mm) for both global $E_{\mathrm{g\text{-}mhpe}}$ and local $E_{\mathrm{mpjpe}}$ (root-relative) settings. We also report acceleration and velocity errors, similar to the object trajectory following the setting but averaged across all body joints. From Table 6, we can see that PHC-X and PULSE-X achieve a high success rate on training data while maintaining a low per-joint error. Distilling from PHC-X to PULSE-X, we observe similar degradation in imitation performance as in PULSE, akin to the reconstruction error in training VAEs [32].

Table 6: Imitation result on dexterous AMASS (14889 sequences).

| | Dexterous AMASS-Train | | | | |
|---|---|---|---|---|---|
| Method | Succ $\uparrow$ | $E_{\mathrm{g\text{-}mpjpe}} \downarrow$ | $E_{\mathrm{mpjpe}} \downarrow$ | $E_{\mathrm{acc}} \downarrow$ | $E_{\mathrm{vel}} \downarrow$ |
| PHC-X | 99.9 % | 29.4 | 31.0 | 4.1 | 5.1 |
| PULSE-X | 99.5 % | 42.9 | 46.4 | 4.6 | 6.7 |

# C  Details about Omnigrasp

## C.1  Object Processing

Since the simulator requires convex objects for simulation, we use the built-in v-hacd function to decompose the meshes into convex geometries. The parameters we use for decomposition can be found in Table 7. To compute object latent code, we use 512-d BPS [58] by randomly sampling 512 points on a unit sphere and calculating their distances to points on the object mesh. As some object meshes have a large number of vertices, we also perform quadratic decimation on the mesh if it contains more than 50000 vertices.

## C.2  Training Details

**Early Termination**. During training, we terminate the episode whenever the object is more than 12cm away from its desired reference trajectory at time step t: $\|\hat{\boldsymbol{p}}_t^{\mathrm{obj}} - \boldsymbol{p}_t^{\mathrm{obj}}\|_2 > 0.12$.

**Table Removal**. Since the GRAB and OakInk datasets are table-top objects, we use a table at the beginning of the episode to support the object. However, since our randomly generated trajectory can collide with the table and the humanoid has no environmental awareness except for the object, we remove the table after certain timestamps (1.5s) during training.

**Contact Detection**. As IsaacGym does not provide easy access to contact labels and only provides contact forces, there is no way of differentiating between contact with the table, humanoid body, or objects. Thus, we resort to a heuristic-based way to detect contact. Specifically, if the object is within 0.2m from the hands, has non-zero contact forces, and has a non-zero velocity, we deem it to have contact with the hands.

**Trajectory Generator**. Randomly generated trajectories can be seen on our supplement site on the OakInk and OMOMO dataset, as there is no paired MoCap object motion for these datasets. We sample a random velocity and delta angle at each time step and aggregate the velocities to produce full trajectories. We bound the velocity of our randomly generated trajectories to be between [0, 2] m/s and bound the angles to be between [0, 1] radian. With a probability of 0.2, a sharp turn could happen where the angle is between [0, $2\pi$]. As the trajectories can not be too high or low, we bound the z-direction translation to be between [0.1, 2.0]. For orientation, we sample a random ending orientation at the end of the trajectory and interpolate it between the object's initial trajectory to obtain a sequence of target rotations.

Table 8: Additional ablations: Object-latent refers to whether to provide the object shape latent code $\boldsymbol{\sigma}^{\text{obj}}$ to the policy. RNN refers to either using an RNN-based policy or an MLP-based policy. Im-obs refers to whether to provide the policy with ground truth full-body pose $\hat{\boldsymbol{q}}_{t+1}$ as input.

| | | | | GRAB-Goal-Test (Cross-Object, 140 sequences, 5 unseen objects) | | | | | | |
|---|---|---|---|---|---|---|---|---|---|---|
| idx | Object Latent | RNN | Im-obs | $\text{Succ}_{\text{grasp}} \uparrow$ | $\text{Succ}_{\text{traj}} \uparrow$ | TTR $\uparrow$ | $E_{\text{pos}} \downarrow$ | $E_{\text{rot}} \downarrow$ | $E_{\text{acc}} \downarrow$ | $E_{\text{vel}} \downarrow$ |
| 1 | ✗ | ✓ | ✗ | **100%** | 93.2% | **99.8%** | **28.7** | 1.3 | 6.1 | 5.1 |
| 2 | ✓ | ✗ | ✗ | 99.9% | 89.6% | 99.0% | 33.4 | 1.2 | 4.5 | 4.4 |
| 3 | ✓ | ✓ | ✓ | 95.2 | 77.8% | 97.9% | 32.2 | 0.9 | **3.2** | **3.9** |
| 4 | ✓ | ✓ | ✗ | **100%** | **94.1%** | 99.6% | 30.2 | **0.9** | 5.4 | 4.7 |

## C.3 Additional Ablations

In Table 8, we provide additional ablations left out due to space limitations. Comparing Row 1 (R1) and R4, we can see that on the GRAB dataset cross-object test set, a policy trained without the object shape latent code $\boldsymbol{\sigma}^{\text{obj}}$ can be on par with a policy with access to it. This is because the humanoid learned a general "grasping" for small objects, and the 5 testing objects do not deviate too much from these strategies. Also, upon inspection, R1 learns to rely on bi-manual manipulation and using two hands when it cannot pick it up with one hand, at which point the object shape no longer affects the grasping pose as much. As a result, R1 suffers a higher rotation error $E_{\text{rot}}$. On the GRAB cross-subject test (44 objects), R1 has a trajectory success rate of $\text{Succ}_{\text{traj}}$ 84.2%, worse than R4's 90.5%. R2 vs. R4 shows that the RNN policy is more effective than the MLP-based policy, confirming our intuition that some form of memory is beneficial for a sequential task, such as grasping and omnidirectional trajectory following. R3 studies the scenario where we provide ground truth full-body pose $\hat{\boldsymbol{q}}_t$ to the policy at all times, similar to the setting in PhysHOI [78] (though without the contact graph). Results show that this strategy leads to worse performance, and also prevents us from training on objects that do not have paired MoCap full-body motion. This indicates that the contact graph is needed to imitate human-object interaction precisely. Omnigrasp provides a flexible interface to support learning and testing on novel objects without needing paired ground-truth full-body motion.

## C.4 Per-object Successrate breakdown

In Table 9, we break down the per-object success rate on the cross-object split of the GRAB dataset. Of the 5 novel objects, our model finds it hardest to pick up the toothpaste, which has an elongated surface. Upon inspection, we find that Omnigrasp will slip on the round edges of the toothpaste surface and fail to grasp the object. Compared to previous SOTA [6], Omnigrasp outperforms in all metrics and objects.

Table 9: Per-object breakdown on the GRAB-Goal (cross-object) split.

| Object | Braun *et al.* [6] | | | Omnigrasp | | |
|---|---|---|---|---|---|---|
| | $\text{Succ}_{\text{grasp}} \uparrow$ | $\text{Succ}_{\text{traj}} \uparrow$ | TTR $\uparrow$ | $\text{Succ}_{\text{grasp}} \uparrow$ | $\text{Succ}_{\text{traj}} \uparrow$ | TTR $\uparrow$ |
| Apple | 95% | - | 91% | **100%** | 99.6% | **99.9%** |
| Binoculars | 54% | - | 83% | **100%** | 90.5% | **99.6%** |
| Camera | 95% | - | 85% | **100%** | 97.7% | **99.7%** |
| Mug | 89% | - | 74% | **100%** | 97.3% | **99.8%** |
| Toothpaste | 64% | - | 94% | **100%** | 80.9% | **99.0%** |

# D Additional Discussions

## D.1 Alternatives to PULSE-X

One alternative way for reusing the motor skills from a motion imitator like PHC-X is to train a kinematic motion latent space to provide reference motion to drive PHC-X. Such a general-purpose kinematic latent space has been used in physics-based control for pose estimation [77] and animation [100]. However, few have been extended to include dexterous hands. These latent spaces, like HuMoR [64], model motion transition using an encoder $\boldsymbol{q}_\phi(\boldsymbol{z}_t|\hat{\boldsymbol{q}}_t, \hat{\boldsymbol{q}}_{t-1})$ and decoder $p_\theta(\hat{\boldsymbol{q}}_t|\boldsymbol{z}_t, \hat{\boldsymbol{q}}_{t-1})$ where $\hat{\boldsymbol{q}}_t$ is the pose at time step t and $\boldsymbol{z}_t$ is the latent code. $\boldsymbol{q}_\phi$ and $\boldsymbol{p}_\theta$ are trained using supervised learning. The issue with applying such a latent space to simulated humanoid control is twofold:

- The output $\hat{\boldsymbol{q}}_t$ of the VAE model, while representing natural human motion, does not model the PD-target (action) space required to maintain balance. This is shown in prior art [77, 100], where an additional motion imitator is still needed to actuate the humanoid by imitating $\hat{\boldsymbol{q}}_t$ instead of using $\hat{\boldsymbol{q}}_t$ as policy output (PD-target).

- $\boldsymbol{q}_\phi$ and $\boldsymbol{p}_\theta$ are optimized using MoCap data, whose $\hat{\boldsymbol{q}}_t$ values are computed using ground truth motion and finite difference (for velocities). As a result, $\boldsymbol{q}_\phi$ and $\boldsymbol{p}_\theta$ handle noisy humanoid states from simulation poorly. Thus, [77] runs the kinematic latent space in an open-loop auto-regressive fashion without feedback from physics simulation (*e.g.* using $\hat{\boldsymbol{q}}_{t-1}$ from the previous time step's output rather

than from simulation). The lack of feedback from physics simulation leads to floating and unnatural artifacts [77], and the imitator heavily relies on residual force control to maintain stability.

## E    Broader social impact.

Our method can be used to create a realistic grasping policy for humanoids, generate animation, or synthesize stable grasps. While the state designs have access to privileged information, the overall system design methodology (plus sim-to-real transfer techniques such as domain randomization) has the potential to be transferred to a real humanoid robot. Thus, it has a potential positive social impact, as it can create content or help build the next generation of home robots.

