# OpenReview forum: "Omnigrasp: Grasping Diverse Objects with Simulated Humanoids"
_NeurIPS.cc/2024/Conference — NeurIPS 2024 poster_

### Official Review · Reviewer_z5bR · 2024-07-11

**Soundness:** 3
**Presentation:** 4
**Contribution:** 4
**Rating:** 7
**Confidence:** 4

**Summary:**

The paper introduces Omnigrasp, a method for controlling a simulated humanoid with dexterous hands to grasp and manipulate a wide variety of objects along complex trajectories. The authors leverage a universal humanoid motion representation to improve training efficiency and scalability. This method achieves state-of-the-art success rates in object manipulation tasks and generalizes well to unseen objects and trajectories. Key contributions include a dexterous motion representation, the allowance for simple state and reward designs for training, and high success rates in diverse object manipulation scenarios.

**Strengths:**

The challenging yet important topic regarding

1. a full-body dexterous grasping, which considers the physical constraints and unstability of the body;
2. omnidirectional movement after grasping, i.e., moving the object along any directions within a reachable range;

is really worth the exploration. The paper not only presents an effective algorithm that does not heavily rely on existing object trajectories but also conducts experiments on extensive objects to evaluate grasping success rates.

**Weaknesses:**

While the paper marks a significant advancement in humanoid dexterous grasping, some inherent flaws in the setting hinder its seamless transfer to real-world scenarios. I.e., some state variables, such as the object mesh, pose, and velocity, cannot be accurately accessed in the real world. This discrepancy between simulation and reality poses a challenge for the practical deployment of the proposed method.

**Questions:**

1. How do you plan to bridge the gap between simulation and real-world applications? Given that the input assumes access to the object pose, velocity, and mesh, which are impossible to obtain in the real world.
2. How do you envision the way your method could be adapted for, or at least benefit, downstream tasks? E.g., more fine-grained dexterous manipulation.

**Limitations:**

1. The approach is validated only in a simulated environment, which may not capture all real-world complexities and variabilities.
2. The method assumes accurate state estimation for object meshes, poses, and velocities, which is challenging to achieve in real-world scenarios.
3. Fine-grained dexterous manipulation is yet to be achieved by this method.

---

> ### Author Rebuttal · Authors · 2024-08-06
>
> Thank you for your helpful comments! We have revised the paper to provide more discussion for sim-to-real transfer and downstream tasks. To address your concerns and questions:
>
> ---
>
> > **Transfer to Real Humanoid**
>
> We acknowledge that Omnigrasp in its current form could not be applied to the real world. Its potential for real-world deployment lies in the scalable learning framework (first learn motion representation and then train grasping and object trajectory following). To deploy in the real-world and avoid using privileged information, the popular practice in sim-to-real is to distill a teacher policy into a student policy that does not use privileged information [1, 2, 3]. We imagine an input space similar to [3] (8 point bounding-box) could be used in the real-world for grasping. The mesh information could be replaced with vision or point cloud [1]. Similar to the transfer observed from the motion imitation task to real humanoid motion imitation [4, 5, 6], we believe that a framework similar to Omnigrasp could be deployed to the real world.
>
> > **More Downstream Task Like Fine-grained Manipulation**
>
> We believe that if we train *specifically* for the object manipulation task for one object, our framework is capable of learning fine-grained manipulation. Our early results indicate that we can *overfit* to fine-grained object movement and handovers using the same motion latent space. This indicates that the motor skills learned in PULSE-X could support fine-grained manipulation. Once we add more diverse objects and trajectories, the learning problem becomes significantly harder and we observe the policy struggle with more precise trajectory tracking. While we focus on scaling the grasping task using simulated humanoids, we believe that tuning the reward for the fine-grained manipulation task on a smaller number of objects can lead to promising results. How to learn a general policy that can do both is an interesting yet challenging future direction.
>
>
> **References**
>
> > [1] DexPoint: Generalizable Point Cloud Reinforcement Learning for Sim-to-Real Dexterous Manipulation
> [2] Robot parkour learning
> [3] DextrAH-G: Pixels-to-Action Dexterous Arm-Hand Grasping with Geometric Fabrics
> [4] Expressive whole-body control for humanoid robots
> [5] HumanPlus: Humanoid Shadowing and Imitation from Humans
> [6] Learning human-to-humanoid real-time whole-body teleoperation

---

> > ### Comment · Reviewer_z5bR · 2024-08-08
> > **Keep my original recommendation**
> >
> > Despite its current limitations in transferability and scalability, I believe this work can provide a foundation for future humanoid grasp learning methods. I will keep my original decision.

---

> > > ### Author Response · Authors · 2024-08-10
> > > **Thank you for your comments and response!**
> > >
> > > The authors appreciate your suggestions and Accept recommendation. Please let us know if you have any additional questions!

---

### Official Review · Reviewer_gXSs · 2024-07-13

**Soundness:** 2
**Presentation:** 3
**Contribution:** 2
**Rating:** 5
**Confidence:** 5

**Summary:**

The paper introduces a method for controlling a simulated humanoid robot to grasp and move objects along a trajectory with the use of a dex hand, This approach enables the robot to handle diverse objects with diverse trajectories. The key contribution is a humanoid motion representation that enhances training efficiency and human-like manipulation skills and applies it to grasping tasks. The method achieves a reasonable success rate in completing object trajectories and generalizes well to unseen objects.

**Strengths:**

* This work extends previous research on PHC with dexterous hand capabilities, enabling whole-body manipulation in simulation with a high success rate. The results, including supplemental videos, demonstrate human-like motion for object grasping and trajectory following.
* Experiments show that the proposed controller exhibits reasonable robustness and generalization ability across objects of different scales.
* Extensive ablation studies validate the effectiveness of various components, providing plausible analysis and explanations for comparisons, such as why experiments without object shape still achieve reasonable grasping performance.
* The paper is well-written, easy to follow, and provides sufficient technical details, including supplemental code, for the community to reproduce the results.

**Weaknesses:**

* Despite extensive ablation studies, the work lacks some important baselines, such as fully end-to-end RL with similar compute (R1 in ablation), considering that training PHC-X and PULSE-X takes half a month. Baselines utilizing human motion prior data in different ways, such as AMP or ASE, are also missing.
* More details about the dexterous hand should be included, given its significance over previous work on PHC. Comparisons with experiments using PHC but not PHC-X, such as using PHC for body control and raw action space for the hand, are needed. The ablation study compares with motion prior to training without Dex-AMASS, but it’s unclear how the hand joints are controlled, and more details are necessary.
* [Minor - Nothing significant]Though the authors claim the work can be extended to real robots in supplementary material, it seems unlikely due to the low simulation frequency (leading to low simulation fidelity), unrealistic robot methodology design, and the use of privileged information.

**Questions:**

* Considering most of the human body motions and hand motions are randomly paired from different datasets, why not train the model separately (PHC for body motion and another model for hand motion)? What additional benefits does modeling them together provide?

**Limitations:**

The authors include a reasonable discussion of the limitation and no obvious potential negative societal impact needs to be discussed.

---

> ### Author Rebuttal · Authors · 2024-08-06
>
> Thank you for your insightful suggestions and feedback. We have revised the paper to provide additional baselines (AMP and PHC), add details about dexterous hands, and discuss decoupled body and hand prior. To address your questions:
>
> ---
>
> > **Additional Baselines**
>
> In Table 2 of the global PDF, Row 2, we train the policy without PULSE-X to collect $10^{10}$ samples for 1 month (Omnigrasp uses $10^{9}$) and observe its performance still significantly lags behind Omnigrasp. Training with only RL (even with AMP) also leads to non-human-like motion (see supplement videos). Row 3 reports training our method with AMP but not PULSE-X, which does not lead to a high success rate. Row 4 reports results from Braun et al. [1], which uses ASE-style latent space **trained specifically for the grasping task** on GRAB data. Compared to Braun et al., we achieve a much higher success rate in terms of grasping and trajectory following and can follow complex trajectories. In comparison to ASE, PULSE-X's latent space is trained on AMASS and covers much broader types of human motion. ASE's discriminative latent space, while effective when learning from specialized and curated datasets, struggles to learn from large-scale unstructured datasets such as AMASS, as shown in [2].
>
> > **Details about the Dexterous Hand**
>
> We will add more details about how the hands are controlled in the revised manuscript. The hands are treated the same as the rest of the body (e.g. toe or wrist) and actuated using PD controllers (Section 4.1).
>
> Using PHC for body control and raw action space for the hand is an excellent suggestion. However, this approach requires we learn a policy to output kinematic poses as "actions" for PHC to imitate. As observed in [2], kinematic motion is a poor sampling space for RL due to its high dimension and lack of physical constraints (e.g. a small change in root position can lead to a large jump in motion). Thus, prior art that uses kinematic motion as the action space (e.g. kinematic policy [3, 4]) uses supervised learning to train the motion generator instead of RL. Supervised learning would require paired full-body human grasping data, which is scarce and limited in diversity. One of the main advantages of Omnigrasp is its ability to learn grasping policy **without paired full-body grasping data**, enabling it to scale to many objects and diverse trajectories.
>
> Further, even if a policy could output ground-truth kinematic pose for PHC, small errors in imitation can lead to the hand missing the objects. To demonstrate this point, we use **ground truth** MoCap as input to a pretrained PHC-X policy (\~30mm imitation error on average) for grasping, using sequences from GRAB. The result from Table 2 (global PDF) Row 1 indicates that the accuracy of a trained imitator does not support object grasping. To use PHC for the grasping task, we will need to fine-tune PHC with object awareness and pair it with a strong kinematic motion generator. Such an approach has been explored for box loco-manipulation [6] without hands, but it only supports moving boxes for now.
>
>
> > **Training with Dex-AMASS**
>
> We apologize for the confusion. When comparing training with or without Dex-AMASS for PULSE-X, the only difference is the **training data**. In both cases, the hand joints are controlled using PD controllers to output PD targets, but one with regular AMASS and one with our Dex-AMASS as training data. We will further clarify this.
>
>
> > **Extending to Real Robots**
>
> We will revise the sentence to "While the state has access to privileged information and the current humanoid has no real-world counterpart, the overall system design methodology has the potential to be transferred to a real humanoid robot, similar to how the motion imitation task is applied in recent humanoid work [5, 7, 8]." We believe that the framework of first learning a universal motion representation, then learning grasping policy in simulation, followed by conducting sim-to-real modifications (e.g. domain randomization, distilling into a vision-based policy), **has the potential** to be applied to real-world humanoids.
>
> > **Train Separate Body and Hand Models**
>
> As mentioned in Section 6, while we utilize a simple yet effective unified motion latent space, separate motion representation for hands and body could lead to further improvements. We completely agree that training a separate model *could* be beneficial. Braun et al [1] use decoupled body and hand prior, and achieve a lower success rate. We hypothesize that since the hand is tethered to the body and performs actions based on different wrist movements (which affects gravity), training a hand-only model that performs well when combined with the body is non-trivial. Since we observe accurate hand-tracking results when training a motion imitator jointly for hands and body, we proceed with the joint latent space design and find it is sufficient for achieving good grasping results. We are actively exploring using decoupled hand and body latent space, though it has not yet shown real benefit.
>
>
> **References**
>
> > [1] Physically plausible full-body hand-object interaction synthesis
> [2] Universal humanoid motion representations for physics-based control
> [3] Learning predict-and-simulate policies from unorganized human motion data
> [4] Dynamics-regulated kinematic policy for egocentric pose estimation
> [5] Learning human-to-humanoid real-time whole-body teleoperation
> [6] Hierarchical planning and control for box loco-manipulation
> [7] Expressive whole-body control for humanoid robots
> [8] HumanPlus: Humanoid Shadowing and Imitation from Humans

---

> > ### Comment · Reviewer_gXSs · 2024-08-13
> >
> > Thank you for addressing the concerns. I'm keeping my original evaluation.
> >
> > One additional question about separately controlling hand and body, though it might require output kinematic pose using PHC, it seems using PULSE (which if I understand it correctly takes latent command as input) + separate controller for hand should address most of the problems authors raised for using kinematic poses. Is there any specific challenge or problem with this?

---

> ### Author Response · Authors · 2024-08-13
> **Follow up to Comment by Reviewer gXSs**
>
> Thank you for the discussion!
>
> PULSE is not compatible with the body of the SMPL-X humanoid, since it is trained using the SMPL humanoid. As the body shape for SMPL and SMPL-X is not the same, SMPL-X and SMPL humanoids have slightly different bone lengths and directions. Since the GRAB dataset we are using is in SMPL-X format, we initiated this work to support the SMPL-X humanoid. As more and more dataset is captured using SMPL-X instead of SMPL-H, we hope our pipelines can be future-proof and support SMPL-X based datasets.
>
> We would also like to note that raw hand actions are problematic as the controller can use non-human-like strategies in grasping (see Supplement Site, Training Without PULSE-X).  Using raw actions leads to hands being distorted due to their high degree-of-freedom and unbounded exploration. To provide motion prior to the hand's motion, either AMP or PULSE-like motion prior will be needed. Using PULSE for the body and then the raw-action space for the hand plus some form of hand prior would also be quite close to our method in terms of methodology and compute overhead (for training PHC and PULSE), and would support our story that using a strong motion prior one can enable learning diverse grasping and object trajectory following. We are actively exploring using separate hand and body priors, though it does not seem to be the bottleneck yet.
>
> Please feel free to ask us any additional questions!

---

### Official Review · Reviewer_rKeN · 2024-07-14

**Soundness:** 3
**Presentation:** 3
**Contribution:** 3
**Rating:** 6
**Confidence:** 4

**Summary:**

This paper proposes an approach to learning a humanoid controller that can manipulate objects to follow trajectories. It first assembles a dataset of human bodies and hands motions, and learns a control policy from the state transitions in the dataset. Then, they distill this policy using a VAE to obtain an action decoder that models realistic human action distributions. After that, they use this distribution as the action space to learn an RL policy whose reward is based on object trajectory following. With this action space, the neural network policy can output realistic actions and avoid exploration difficulties. The authors demonstrate several impressive, yet natural and physically plausible motions for a simulated humanoid to grasp and move objects according to trajectories.

**Strengths:**

The results are generally good, and the motion is relatively natural and physically plausible.

The demonstration of extending this approach to high-dimensional control is important.

The authors also provide several insightful observations, such as how to assemble the dataset to extend PULSE to human hand motion and how to achieve diverse grasping behaviors. These contributions are all important to the community.

**Weaknesses:**

While I definitely think this paper is good, I believe there are several critical aspects that should be more carefully evaluated, either qualitatively or quantitatively. I will provide the summary comments in this section and write the specific questions that I would like answers to during the discussion phase in the next section.

First, I’m generally curious about the robustness of the proposed system. The object trajectory following policy takes a few inputs without noise from the simulator (joint positions, object latent code, proprioception). However, in reality, these quantities are far from perfect.

Second, this approach assumes the humanoid is similar enough to the mocap data. I agree this is a reasonable assumption. However, in reality, humanoid robots and their hands are usually not perfect replicas of humans. How similar do their morphologies need to be?

Regarding the learning approach, it first learns a policy to output actions given the human sequence data. Then it distills this policy network using VAE to learn the distributions as the action space. There should be an alternative approach that can serve as a simple baseline (details below). This is not discussed (or I missed the reference?).

The current reward contains several terms with different reward weights and hyper-parameters. I would not say this is a simple reward, especially without a concrete comparison to previous work.

**Questions:**

Regarding the robustness of the system: How would the policy perform in noisy situations? For example, when the joint positions are not accurate (real robots will have imperfect joint encoders). And how accurate does the object mesh need to be?

How important is the morphology similarity between the humanoid and the hand? For example, do they need to have the same number of joints or links? I also observed even in the current system, it seems the simulated humanoid does not perfectly match human kinematics. Is this correct? If so, how do you bridge this gap?

Regarding the simple baseline alternatives:
* Instead of learning actions from human trajectory sequences between two timesteps, why not directly using a VAE to reconstruct human poses? This VAE will directly learning a distribution of human joint positions. Then in the RL policy training, you can map the policy output to natural human joint positions using this VAE.
* Even when one wants to learn the “actions” instead of “joint positions”, one could use a encoder-decoder structure (either VAE or vanilla auto-encoder) as the policy backbone and directly learn in the PULSE-X phase, instead of separating it into two stages.
I’m curious if the authors tried these alternatives before? Or does this comparison exist in the literature? Are they performing not well? It would be much helpful to analyze this problem or provide references in the text.

A more comprehensive discussion on the failure cases would be helpful. You show several videos of the failure cases, which I think is very helpful. But why do they fail? Is it because the objects themselves are quite difficult? Regarding this point, it would also be much helpful to have a per-object accuracy analysis.

**Limitations:**

The authors describe the limitations mainly as not performing dexterous in-hand manipulation. The authors could consider using recent learning-based dexterous in-hand manipulation work as the “mocap dataset” in your formulation and learn action distribution using it.

I think there is another limitation, which is that the current formulation does not demonstrate capability when the object needs to interact with the environment.

While I acknowledge that the above two points are out of scope and will not affect my assessment at all, they might be interesting future directions.

---

> ### Author Rebuttal · Authors · 2024-08-05
>
> Thank you for the helpful feedback and comments. We have revised the paper to include robustness tests, a discussion about kinematic latent space, and failure case analysis. To address your concerns:
>
> ---
>
> > **Robustness**
>
> In Table 1 of the global PDF, we add uniform random noise [-0.01, 0.01] to both task observation $s_t^g$ (positions, object latent codes, etc.) and proprioception $s_t^p$. A similar scale (0.01) of random noise is used in sim-to-real RL to tackle noisy input in real-world humanoids [1]. We see that Omnigrasp is relatively robust to input noise, even though it has not been trained with noisy input. Performance drop is more prominent in the acceleration $E_{\text{acc}}$ and velocity $E_{\text{vel}}$ metrics. Adding noise during training can further improve robustness. We do not claim that Omnigrasp is currently ready for real-world deployment, but we believe that a similar system design **plus sim-to-real modifications** (e.g. domain randomization, distilling into a vision-based policy) has such potential.
>
> > **Morphology**
>
> The AMASS [2] dataset has been successfully used in real-world humanoid control [1, 3, 4]. Retargeting techniques can be used to transfer human motion to the humanoid, even when they do not have the same morphology or joint numbers. Similarly, human hand motion in Mano [6] format can be transferred to robotic hands [7, 8].
>
> One of the main strengths of Omnigrasp is its independence from paired full-body grasping and object trajectory data, which is expensive to capture and therefore only available in small-scale datasets. Since the retargeting process could introduce errors that cause human-object interaction to be imprecise, retargeted interaction data can be hard to use for methods that rely on demonstrations. Our two-stage process (first obtain motion prior and then learn grasping policy) can leverage imperfectly retargeted motion, since as long as the motion is human-like, we can acquire motor skills by imitating it.
>
>
> > **Baseline Alternative**
>
> Excellent advice! Such a general-purpose kinematic latent space has been used in physics-based control for pose estimation [10] and animation [11], though few have been extended to include dexterous hands. These latent spaces, like HuMoR [9], model motion transition using an encoder $q_\phi (z_t| x_t, x_{t-1})$ and decoder $p_\theta(x_t | z_t, x_{t-1})$ where $x_t$ is the pose at time step t and $z_t$ is the latent code. $q_\phi$ and $p_\theta$ are trained using supervised learning. The issue with applying such a latent space to simulated humanoid control is twofold:
>
> - The output $x_t$ from the VAE model, while representing natural human motion, does not model the PD-target (action) space required to maintain balance. This is shown in prior art [10, 11], where an additional motion imitator is still needed to actuate the humanoid by imitating $x_t$ instead of using $x_t$ as policy output (PD-target).
> - $q_\phi$ and $p_\theta$ are optimized using MoCap data, whose $x_t$ values are computed using ground truth motion and finite difference (for velocities). As a result, $q_\phi$ and $p_\theta$ handle noisy humanoid states from simulation poorly. Thus, [10] runs the kinematic latent space in an open-loop auto-regressive fashion without feedback from physics simulation (e.g. using $x_{t-1}$ from the previous time step's output rather than from simulation). The lack of feedback from physics simulation leads to floating and unnatural artifacts [10], and the imitator heavily relies on residual force control [12] to maintain stability.
>
> PULSE directly models the action distribution instead of the kinematic pose and does not need a motion imitator during inference.  Directly learning the latent space without distillation is provided as an ablation in the PULSE paper (Section C.2 Table 6), and it shows that training using RL does not converge to good performance. Random sampling for the variational bottleneck together with random sampling for RL leads to noisy gradients, which hinders policy learning.
>
> We will add additional discussion to elaborate on these points.
>
> > **Reward Complexity**
>
> We will clarify this in the text. The “simple reward” here refers to not needing paired full-body-and-hand MoCap data in the reward, which increases complexity. Prior art often involves graph-based [13, 14] or style rewards [5] that depend on paired data.
>
> > **Failure Analysis**
>
> We categorize the failures into two types: failure due to grasping and due to dropping the object after during transport. The “restabilizing the object in hand” during transport behavior might require further dexterity and training rewards to master. Appendix Section C.3 provides a per-object breakdown of the GRAB-Goal split. We can see that toothpaste and binoculars have the lowest trajectory following success rates (80.9% and 90.5%). In our experience, objects that can roll (such as toothpaste) and large objects (such as binoculars) are more difficult.
>
> **Reference**
>
> > [1] Learning human-to-humanoid real-time whole-body teleoperation
> [2] AMASS: Archive of motion capture as surface shapes
> [3] Expressive whole-body control for humanoid robots
> [4] HumanPlus: Humanoid Shadowing and Imitation from Humans
> [5] Physically plausible full-body hand-object interaction   synthesis
> [6]  Embodied hands: Modeling and capturing hands and bodies together
> [7] Kinematic Motion Retargeting for Contact-Rich Anthropomorphic Manipulations
> [8]  Task-oriented hand motion retargeting for dexterous manipulation imitation
> [9] Humor: 3d human motion model for robust pose estimation
> [10] Learning human dynamics in autonomous driving scenarios
> [11] Learning Physically Simulated Tennis Skills from Broadcast Videos
> [12] Residual force control for agile human behavior imitation and extended motion synthesis
> [13] Simulation and retargeting of complex multi-character interactions
> [14] Physhoi: Physics-based imitation of dynamic human-object interaction

---

### Author Rebuttal · Authors · 2024-08-06

# General Response

The authors would like to thank the reviewers for their time and constructive feedback. We hope that our responses clarify and address their concerns. We are glad that the reviewers find our work a "significant advance" (z5bR), our results "achieve a high success rate" (gXSs, z5bR), and our motion "impressive, natural, and human-like" (rKeN, gXSs). Here, we briefly address some common questions.

> **Transfer to the Real-world**

We fully acknowledge that the current system cannot be transferred to the real world without modification. In this work, we focus on using *simulated humanoids* to grasp diverse objects (>1200) and follow diverse trajectories, a capability that has yet to be attained in simulation with humanoids. We hope that we are taking a step towards real humanoid capabilities. The privileged state used in Omnigrasp could be replaced with values easier to access in the real world (point cloud, pose estimation, etc.), and the humanoid we use can be replaced with a humanoid robot [1]. By first learning a policy that has access to all the available information that can be provided, we hope that we can enable new capabilities for simulated humanoids and design systems that can be modified for real-world deployment (via sim-to-real transfer, teacher-student distillation, etc.)


> **Provided Global PDF**

In the global PDF, we provide two tables containing additional experiments and baselines. Table 1 demonstrates the robustness of a pretrained Omnigrasp policy under artificial noisy conditions. While this experiment will not show Omnigrasp's performance when deployed in the real world, it demonstrates the system's potential to undergo sim-to-real modification for deployment. Table 2 provides additional baselines including PHC, AMP, and training with only RL.


**References**

> [1] Learning human-to-humanoid real-time whole-body teleoperation

---

### Decision · Program_Chairs · 2024-09-25

**Decision:**

Accept (poster)

**Comment:**

The manuscript presents a novel pipeline, called Omnigrasp, that is able to learn controllers for humanoids with dexterous hands such that they can manipulate a wide range of objects along complex paths. By employing a universal motion representation for humanoids (called PULSE-X), the authors enhance training efficiency and scalability. Omnigrasp demonstrates state-of-the-art success rates in object manipulation tasks and strong generalization to new objects and trajectories.

The authors' responses were clear, concise, and offered valuable discussions. Overall, their comments reflect a strong grasp of the literature. Additionally, all reviewers had a positive opinion of the work. The authors also discussed about the limitations of their approach, and I expect that the text will be enhanced with the discussions/comments by the reviewers. For those reasons, I recommend acceptance (poster).